bioengineering/biophysics/sensory biophysics

infrared sensing, hair anatomy, infrared antennae, guard hair, heat sensors

**Author for correspondence:**
Ian M. Baker
e-mail: ianmbaker@live.com

# Infrared antenna-like structures in mammalian fur

Ian M. Baker

Leonardo UK Ltd, Southampton, Hampshire SO15 0LG, UK

IMB, 0000-0001-6265-9837

Many small animals, including shrews, most rodents and some marsupials, have fur composed of at least four types of hair, all with distinctive and complex anatomy. A ubiquitous and unexplained feature is periodic, internal banding with spacing in the 6–12 µm range that hints at an underlying infrared function. One bristle-like form, called guard hair, has the correct shape and internal periodic patterns to function as an infrared antenna. Optical analysis of guard hair from a wide range of species shows precise tuning to the optimum wavelength for thermal imaging. For heavily predated, nocturnal animals the ability to sense local infrared sources has a clear survival advantage. The tuned antennae, spectral filters and waveguides present in guard hair, all operating at a scale similar to the infrared wavelength, could be a rich source of bio-inspiration in the field of photonics. The tools developed in this work may enable us to understand the other hair types and their evolution.

## 1. Introduction

Mammalian fur provides a range of well-documented functions, including thermal insulation [1], waterproofing, protection from ultraviolet radiation, camouflage for avoiding predation [2] and tactile sensing [3,4]. However, these important functions do not provide a complete explanation for the complex microscopic anatomy of individual hairs. There are many types of hair in mammalian fur and in the case of the house mouse, *Mus musculus*, four have been named: guard (or monotrich), awl, auchene and zigzag hair [5]. Zigzag hair is named after its characteristic angular hair shaft and is the main hair in the underfur. Guard hairs, constituting 1–3% of the pelage, are spear-shaped, straight and usually protrude from the fur. The hair types are starkly different yet many multi-species surveys of mammalian hair [6,7] show that these basic types are present in shrews, most rodents and some marsupials. Studies on the tactile function [8,9] show differences in mechano-receptors and nerve fibres with specific afferent routes to the spinal cord and cortex. It is a recurrent theme that each hair type appears to be highly specialized and the structure–function relationships have not yet been the subject of discussion.

### PUBLISHING

**Figure 1.** Two hair types from *Mus musculus* showing characteristic banding and a scale compatible with an infrared function. The small hairs are so-called zigzag hairs and the large one is the widest part of a spear-shaped guard hair—approximately the same width as a human hair.

Figure 1 shows a photomicrograph of hairs from *Mus musculus* with two zigzag hairs and a much broader guard hair. It illustrates the characteristic dark banding, a ubiquitous feature of the hair of small mammals. The band spacing is in the 6–12 µm range and varies little between species. Our measurements show that the 4 g pigmy shrew, *Sorex minutus*, has similar band spacing to the 2 kg European rabbit, *Oryctolagus cuniculus*. The dimensions are similar to the wavelength of infrared radiation and this was the basis of our efforts to find an infrared interpretation of hair anatomy.

The resilience of modern hair anatomy is supported by the fossil record. The oldest hair specimen trapped in amber was dated at 100 million years ago (Ma) and shows that modern cuticular features were already present in the early Cretaceous [10]. The oldest fossil with identifiable hair structure (guard hair and underfur) is a rat-like animal (*Spinolestes xenarthrosus*) dated at 125 Ma [11]. Remarkably, our studies have shown that rodents and small marsupials have equivalent hair types with almost identical microscopic dimensions, despite an evolutionary split in the early Jurassic, 170 Ma [12]. Figure 2 compares the band spacing and cuticle patterns in zigzag and guard hair of *Mus musculus* and the Australian, mouse-like, marsupial *Antechinus agilis*, the agile antechinus. Zigzag hair has converged on a common solution with a grating-like structure on the distal end. Guard hair has diverged but crucially the band spacing in the distal end and the cuticle periodicity are very similar. The inheritance or convergent evolution of hair anatomy suggests that each hair form provides a vital survival function for the animal. In stark contrast, hair forms that are clearly for other purposes, such as whiskers and auditory hairs, have no periodic internal structure.

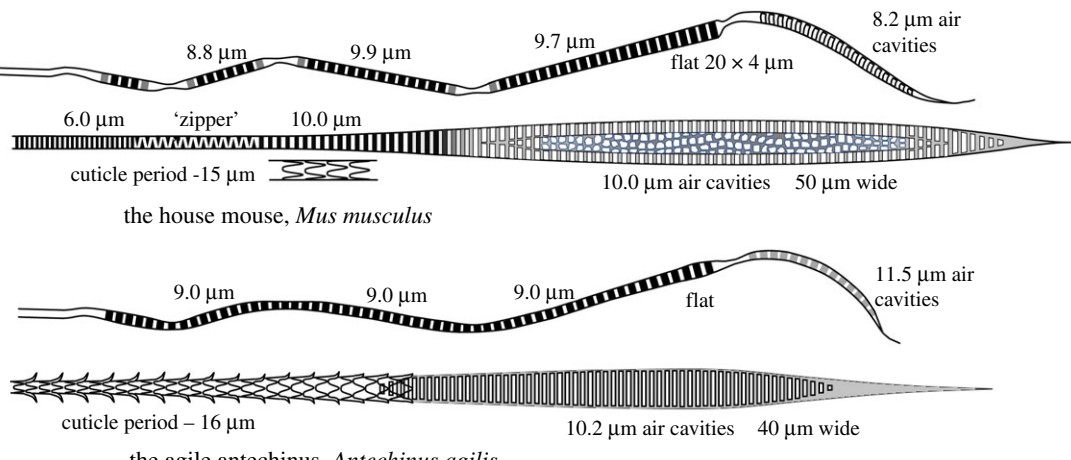

**Figure 2.** Schematics and band spacing measurements on zigzag and guard hairs from the rodent *Mus musculus* and marsupial mouse *Antechinus agilis* showing similar microscopic dimensions despite an evolutionary split now believed to be in the early Jurassic [12].

There are two infrared adaptations that could provide a vital survival benefit for small mammals. The most obvious given the distribution of guard hairs around the body and the vulnerabilities of these species is the ability to sense the infrared signal of a potential predator. This would provide an early warning for species that are often ambushed by stealthy and cryptic predators. The role of guard hairs as the antenna part of an infrared sensor is the main focus of this paper. A second potential adaptation, that applies to small, heavily furred mammals with high metabolic demands, is radiative infrared cooling. The ability to stay cool during an energetic escape from a predator is a clear survival advantage. Many features of zigzag hair support an infrared radiator explanation but there is too much missing science to develop this further at present.

## 2. Infrared antenna interpretation of guard hair

Guard hairs have the key requirements of an antenna. They are stiff, straight, have no rotation or spiralling and are uniformly patterned throughout their length to provide transmission of a dominant wavelength. The tip is very fine and long, and this serves to minimize reflection for on-axis infrared photons, a feature that favours the antenna interpretation of guard hair rather than the tactile one. Figure 2 shows a broader section, often called the shield, that is interpreted as an infrared absorber, a structure that promotes the maximum cross-sectional area for absorption. The lower section is much narrower and promotes the concentration of infrared energy into the base of the hair. Antennae of this type are directional and absorb radiation in the axis of the hair. The fur therefore behaves like the compound eye of an insect to provide all-round infrared sensing.

Optical modelling requires a clear understanding of the physical nature of the dark bands. Through a microscope with back-illumination, the bands are black or brown and appear to focus indicating a higher refractive index. This is consistent with eumelanin, a common pigment in mammalian hair that is known to have a high refractive index [13]. Some arthropods use alternating layers of eumelanin to produce structural colours by a process called Bragg reflection [14]. Eumelanin absorbs strongly at UV and visible wavelengths but becomes transparent at longer infrared wavelengths [15]. The chemical composition of hair is remarkably conserved between species and over time [16] and we speculate that this may be driven by infrared transparency because hair keratin in general reaches peak transparency around 10 µm wavelength [15]. We can therefore model the hair shaft as alternating bands of different refractive indices.

There is an almost identical man-made analogy called a fibre Bragg grating (FBG) used to filter light of certain wavelengths in an optical fibre [17]. Light guided down the fibre experiences weak reflection at each refractive index change and if this reflected light is out-of-phase then it forces the light to propagate through the FBG. This is the principle behind anti-reflection coatings. The transmission condition can be defined by simple formulae,

$$\text{for second order,} \quad d = 0.75\lambda/n \tag{2.1}$$

and

$$\text{for third order,} \quad d = 1.25\lambda/n, \tag{2.2}$$

where $d$ is the band spacing, $\lambda$ is the infrared wavelength and $n$ is the refractive index. The order is the number of wavelengths between adjacent reflections. Hair anatomy appears to mainly employ the second and third order.

The infrared antenna interpretation of guard hair can be tested mathematically because all the morphological detail is bound by first-order optics. Mathematical testing on a range of species is crucial to separate infrared adaptations from coincidental mechanical structure. Specifically, we must show that guard hair is tuned to the optimum wavelength for thermal imaging. Thermal cameras operate at a wavelength of 8–12 µm to exploit a transparent window in the atmosphere and coincide with the peak infrared energy at 10 µm for objects around normal ambient temperatures [18].

## 3. Optical analysis of guard hair

To establish the function of guard hair requires a detailed mapping of the shape, internal banding and cuticular features of the whole hair. We have focused the mapping study on three key species from different Orders: *Mus musculus*, the house mouse, typical of most small rodents, *Antechinus agilis*, the agile antechinus (a mouse-like marsupial) and *Sorex araneus*, the common shrew. Published multi-

species databases have provided supplementary data for a wider range of animals. This paper adopts the nomenclature of Teerink [7] for hair anatomy.

## 3.1. Evidence from the guard hair of rodents (Rodentia) and rabbits (*Leporidae*)

Published databases [7,19] have photographs of guard hair from many species of rodents and rabbits that are very similar in structure. The guard hair of the house mouse, *Mus musculus*, is drawn schematically in figure 3 with typical dimensions. The distal end has the characteristic spear shape (shield) consisting of two tubes with uniform, periodic air cavities, connected by a patterned membrane. In the base of the shield, the air cavities fill with polymer and merge, becoming periodic dark bands in the hair shaft. Lower in the shaft, the bands break into a so-called 'zipper' consisting of dark hemispheres that rotate around the axis, this section being quite variable between hairs. The proximal end of the hair narrows and has a densely banded section with a prominent cuticle pattern. *Mus musculus* has guard hair with remarkably similar microscopic dimensions to the European rabbit *Oryctolagus cuniculus* suggesting that the morphology has changed little since the evolutionary split reported to be in the late Cretaceous [20].

Detailed measurements were made on 70 *Mus musculus* guard hairs from the back of one mouse as described in [21]. With reference to figure 3, the median band spacing each side of the 'zipper' was measured as 10.0 µm and 6.0 µm. For transmission of a dominant wavelength, this must represent an order change; the ratio 0.6 exactly matches the transition from third order ($1.25\lambda/n$) to second order ($0.75\lambda/n$). This establishes that the absorber follows the third-order equation, $\lambda = nd/1.25$, and can be used to determine the tuned wavelength once the refractive index, $n$, is established.

The refractive index of α-keratin at 10 µm can be determined from the Cauchy dispersion equation $n = A + B\lambda^{-2}$, with $A = 1.532$ and $B = 5890$ nm$^2$, giving an extrapolated refractive index for α-keratin of 1.53 at 10 µm [22]. In the top of the hair, the periodic air cavities have the effect of reducing the refractive index. Measurements over many species have found a keratin-to-air ratio averaging 1 : 1.22 providing an effective refractive index of 1.25. Substituting in equation (2.2) gives $d = \lambda$. The tuned wavelength is then simply the measured band spacing in the air cavity section of the hair. There may be an absorption benefit of matching the free-space wavelength to the band spacing since it is present in all the species we have studied. From a sample of 70 *Mus musculus* guard hairs, the median tuned wavelength was 10.0 µm with a standard deviation of 0.30 µm and inter-decile range of 9.4 to 10.5 µm [21]. The Teerink database [7] has well-scaled photographs that can be analysed to extract the band spacing of 16 species of mice, voles and rabbits providing an average band spacing of 9.9 µm (table 1). No exceptions have yet been found. We conclude that a wide range of small mammal species has guard hair with antenna-like features tuned to the infrared window between 8 and 12 µm.

The shield fulfils the Bragg transmission condition for 8–12 µm but there are other wavelengths belonging to different orders that are also transmitted; 8–12 µm radiation has good atmospheric transmission but other wavelengths are absorbed by water vapour in the air and so represent the air temperature rather than the subject of interest. The sensitivity of the antenna is improved if these other wavelengths are radiated away. The 'zipper' and lower hair shaft can be shown to fulfil this

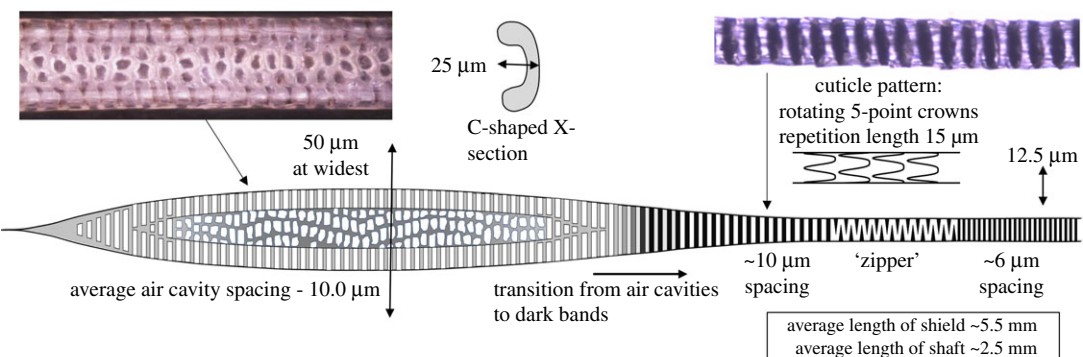

**Figure 3.** A schematic of the guard hair of the house mouse *Mus musculus* is typical on most rodents. The wide section has two rows of regularly spaced air cavities (approximately 600 in total with an average spacing of 10.0 µm) that operates as an absorber, connected by a patterned membrane. The shaft operates as a spectral filter to transmit radiation in the infrared waveband of 8–12 µm and radiate other wavelengths.

**Table 1.** The Teerink database [7] has well-scaled photographs that can be analysed to extract the band spacing (tuned wavelength) for a range of mammals. The data support our detailed measurements that a wide range of species have guard hair tuned to a wavelength range between 8 and 12 μm.

| species | | straight guard hair | | | kinked guard hair | | |
|---|---|---|---|---|---|---|---|
| common | Latin | length | width | scale | length | width | scale |
| bank vole | *Clethrionomys glareolus* | 13 | 69 | 9.0 | 13 | 43 | 8.6 |
| ground vole | *Arvicola terrestris* | 17.5 | 69 | 9.5 | 17.5 | 38 | 8.6 |
| pine vole | *Ondatra zibethicus* | 11.8 | 56 | 10.7 | 11.8 | 46 | 10.1 |
| common vole | *Pitmys subterraneus* | 12.5 | 61 | 11.4 | 12.5 | 43 | 10.1 |
| short-toed vole | *Microtus arvalis* | 11.8 | 51 | 10.7 | 11.8 | 56 | 10.7 |
| root vole | *Microtus agrestis* | 16.3 | 58 | 11.4 | 16.3 | 46 | 11.4 |
| harvest mouse | *Micromys minutis* | 11.3 | 46 | 9.5 | 11.3 | 28 | 9 |
| wood mouse | *Apodemus sylvaticus* | 12 | 38 | 9.0 | 12 | 36 | 9 |
| yellow-necked mouse | *Apodemus flavicolis* | 9.3 | 63 | 9.0 | 9.3 | 51 | 10.1 |
| striped field mouse | *Apodemus agrarius* | 10.8 | 71 | 9.5 | 10.8 | 58 | 9 |
| house mouse | *Mus musculus* | 8.3 | 46 | 10.7 | 8.3 | 41 | 10.1 |
| red squirrel | *Sciurus vulgaris* | 17.5 | 63 | 11.4 | 17.5 | 38 | 10.7 |
| grey squirrel | *Sciurus carolinensis* | 17.5 | 84 | 10.7 | 17.5 | 71 | 8.2 |
| brown hare | *Lepus europaeus* | 28.5 | 69 | 10.7 | 28.8 | 81 | 9.5 |
| blue hair | *Lepus timidus* | 30 | 86 | 8.6 | 30 | 102 | 10.7 |
| rabbit | *Oryctolagus cuniculis* | 32.5 | 69 | 9.5 | 32.5 | 71 | 8.6 |
| | | mm | μm | μm | mm | μm | μm |
| | | | | average | | | average |
| | | | | 10.1 | | | 9.7 |

role. Figure 4a shows a numerical calculation of the infrared spectrum for the antenna with a band spacing of 10 μm showing the unwanted, sideband wavelengths described in detail in [21]. The transmission spectrum is superimposed on the blackbody energy spectrum for 27°C. Figure 4b shows the transmission spectrum for the 'zipper' and lower hair shaft with a band spacing of 6 μm. Note that the peak is still at 10 μm but the sidelobes are shifted. To determine the spectrum of the shield and shaft in combination figure 4a,b are multiplied giving the spectrum in figure 4c. The spectrum

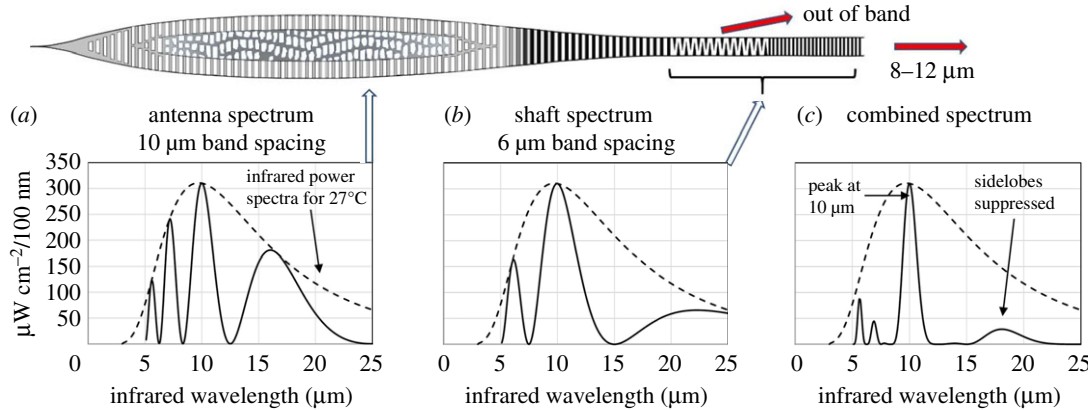

**Figure 4.** An optical analysis of *Mus musculus* guard hair [21] shows the 'zipper' operates as a spectral filter; (a) shows the transmission spectra of the antenna superimposed on the thermal spectrum for 27°C (antenna spectrum 10 μm band spacing); (b) shows the transmission spectra of the lower shaft (shaft spectrum 6 μm band spacing); (c) shows the combined spectra with suppressed sidelobes. The 'zipper' enhances the signal-to-noise for sensing infrared radiation at 8–12 μm.

displays strong sideband suppression. Without the filter, the signal (from the 8–12 µm band) constitutes 33% of the total energy. In the ideal case presented in figure 4c, the spectral filter increases the signal to 72% of the total. The 'zipper' may also preferentially radiate off-axis photons, tightening the reception cone of the antenna. It therefore plays a vital role in enhancing the performance of the antenna. Because the 'zipper' functions as a simple disruptive device to stimulate radiation, it does not have to be geometrically perfect, hence the general variability observed in hair shafts.

The complementary functions of the absorbing antenna and optical filtering show the structure to be a highly integrated optical device. At a top level, the optical performance can be modelled quite well but at the microscopic level the fine detail is far from understood. Since the shield and shaft have wave-guiding structures with dimensions typically less than the wavelength of infrared radiation there could be useful bio-inspiration in the field of photonics.

## 3.2. Evidence from the agile antechinus, *Antechinus agilis*, a mouse-like marsupial

*Antechinus agilis* is a mouse-like marsupial from southeastern Australia (weighing 20 g and measuring 100 mm in length). Figure 5 shows the simplest of three antenna-like hair forms found in the fur of

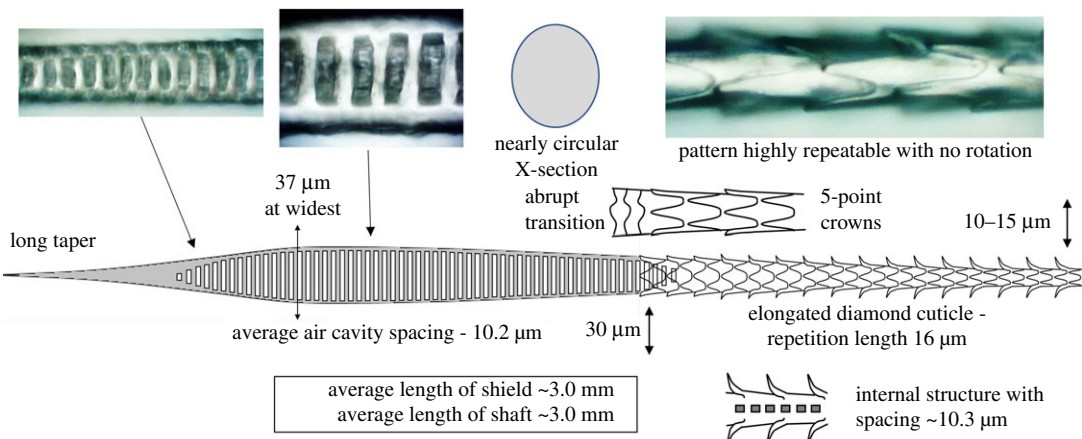

**Figure 5.** Guard hair of *Antechinus agilis*, a mouse-like marsupial from southeastern Australia. The wide distal end has regularly spaced air cavities (250 in total with a spacing of 10.2 µm). The cuticle pattern on the shaft begins abruptly at the start of the shaft and has a wave-like nature and repetition distance indicative of a spectral filtering function. The cuticle pattern has a similar scale to that of *Mus musculus* in figure 3.

*Antechinus agilis.* As expected from the distant relationship to eutherian mammals, the guard hair differs radically in morphology, but the details that relate to infrared adaptations are similar. The hair cross-section is nearly circular, contains around 250 periodic air cavities with an average spacing of 10.2 µm, and this section has no significant cuticle pattern. The topology changes abruptly into a tapered shaft with a very prominent and repeatable cuticle pattern, especially at the base.

*Antechinus agilis* guard hair is an outstanding subject for further study as it is symmetric and cleanly divided into a simple antenna and a tapering shaft. The heavily patterned, tapering shaft is of particular interest for photonic bio-inspiration. In §3.1, the features in the hair shaft of rodents were shown to perform a spectral filtering function and tighten the reception angle of the antenna. By analogy, the antechinus hair shaft may perform the same function by using the so-called elongated diamond pattern [7]. A tapered waveguide normally results in radiation after a few reflections. The cuticle pattern has a constant repetition distance of 16 µm (1.5× the tuned wavelength of 10.2 µm) indicating its purpose is to frustrate the radiation of 10 µm radiation, while other wavelengths are radiated away. The mathematical confirmation of this function is not available now but should be possible in future with the growing interest in computer-generated surface holograms [23]. *Mus musculus* has a very similar cuticle pattern with identical repetition distance but the crowns rotate around the shaft so it is more difficult to study than *Antechinus agilis*

## 3.3. Evidence from antenna-like hairs in shrews

Whisker-like guard hairs are reported in shrew fur [7] but our samples lacked them. Instead, the predominant sensor-like hair in shrews has a compact antenna and a zigzag shaft with each bend

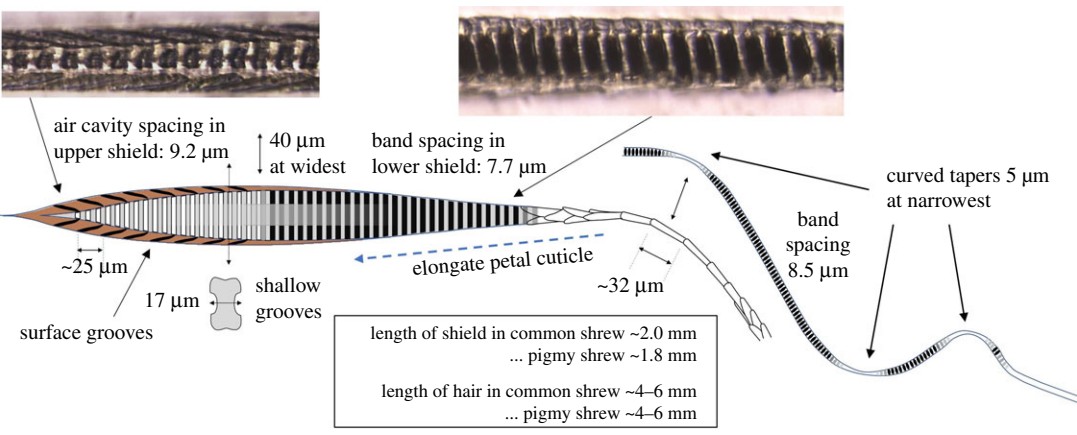

**Figure 6.** The infrared sensor-like hair in the common shrew, *Sorex araneus*, has a different concept to rodents consisting of a complex and sophisticated-looking antenna on a zigzag shaft. The antenna is crucially tuned to a single wavelength. The shaft has three or four tapered bends in different directions that may be performing the same spectral filter function as the shaft of *Mus musculus* in figures 3 and 4.

projecting in a different direction. Shrews provide an opportunity to test if the antenna is tuned to a single wavelength. The analysis concentrated on the common shrew, *Sorex araneus*, (weighing 10 g and measuring 80 mm in length) supported by data on the pigmy shrew, *Sorex minutus*, (weighing 4 g and measuring 55 mm in length). The antenna-like hairs of these species are very similar, differing slightly in length but not in proportion to the body size. Figure 6 shows the hair anatomy of *Sorex araneus*.

With reference to figure 6, the band spacing in the distal, air cavity section is 9.2 μm. A key feature is the change from air cavities to a solid polymer section at the widest part of the hair. The effective refractive index should change from 1.25 for the air cavity section (as described in §3.1) to a higher value in the solid part. If the structure is tuned to a single wavelength, the band spacing must adjust to accommodate the change in refractive index. Figure 6 shows that there is indeed a band-spacing change and it corresponds to a refractive index of 1.50 in the proximal end, a value close to the 1.53 derived in §3.1. The confirmation that the band spacing is compatible with a single wavelength is strong evidence to support the antenna model because there is no alternative explanation for this structural accommodation. The measurement techniques and data are presented in [21].

The *Sorex araneus* hair drawn schematically in figure 6 has a zigzag shaft and in common with other zigzag hairs each angle projects in a different direction so from the top they project to the N, S, E and W. The narrow parts are fragile and there must be a strong selection pressure (e.g. an optical function) to overcome the mechanical weakness this feature introduces into the hair. We speculate that this may be a strategy for radiating unwanted photons covering all polarization angles, much as rodents and rabbits employ the 'zipper' structure in the shaft. The shrew family appear to have short but sophisticated antennae with many detailed features that are yet to be explained.

# 4. Non-photonic evidence for the infrared interpretation

## 4.1. Mammals that lack antenna-like guard hairs

Bats lack antenna-like guard hairs, probably because flight does not allow them to dwell long enough on the scene to collect enough signal. They have a type of guard hair that is modified to wrap over the underfur and present an aerodynamic shape, but it lacks antenna-like features. Notably, the European mole, *Talpa europeae*, lacks antenna-like guard hair, presumably because it has no need for 360-degree infrared threat warning in underground tunnels. Moles also have a type of guard hair to protect the underfur but these are modified underfur hairs with no antenna-like features. We have not found any photographic evidence in the literature of infrared sensors in animals larger than a rabbit or in small ground-based predators, such as weasels. Clearly, a wide species survey is required to confirm this but sensor-like guard hairs appear to be restricted to heavily predated animals with a need for infrared vigilance.

## 4.2. Potential counter-adaptations in predators of small rodents

If the infrared sensor interpretation of guard hair in small rodents is valid, there should be counter-adaptations in their common predators. A Leonardo Merlin camera was used to look for evidence of reduced infrared brightness or infrared concealment adaptations. Although only indicative at this stage there are some striking observations. Most warm-blooded animals appear very bright in thermal cameras but snakes, small cats and owls appear to be exceptions. Snakes in vegetation are virtually invisible in the thermal infrared, even during movement, so they are very effective ambush predators. The domestic cat, when hunting, has very weak infrared emission from the cold nose region and suppressed emission in general (figure 7a). In the stalking pose, cats project the cold nose forward

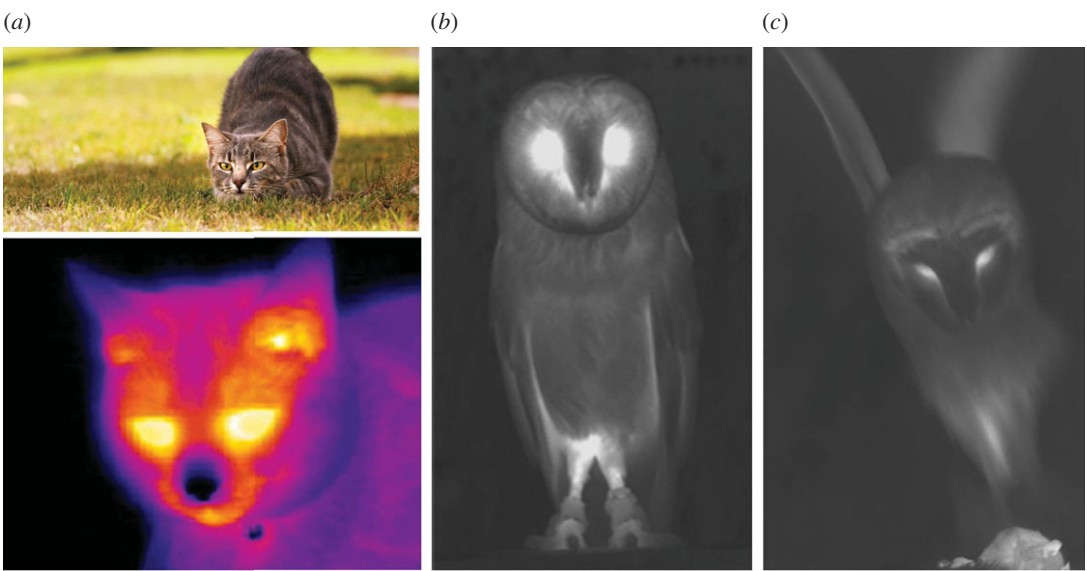

**Figure 7.** (a) Is a typical thermal image of a domestic cat with a characteristic cold nose that is projected towards the prey during the stalking pose; (b) illustrates the cold centre of the facial disc of a barn owl, *Tyto alba,* and low emission from the feathered areas in general; (c) is a late frame from a video showing a barn owl diving onto a mouse with suppressed infrared emission in the direction of the prey.

effectively compensating thermal emission from the eyes (which are in any case squinted). The common barn owl, *Tyto alba*, has a cold centre stripe in the facial disc equivalent to the cold nose of a cat (figure 7b). A common barn owl was filmed diving on a mouse with the Leonardo Merlin camera set to an exposure of 1 ms. Figure 7c shows one frame of the video. When flying, owls have strong infrared emission from the armpit area, but this is hidden behind the body during a dive. Thermal emission from the rest of the body is strongly suppressed by absorption in the feathers including the upper leg feathering. The overall result is a very low thermal signal presented to the prey. The full video can be viewed at [21].

# 5. Discussion on infrared sensitivity and the potential thermoreceptor

Thermal infrared radiation is due to a continuous energy interchange between vibrating atoms and electromagnetic waves. At normal ambient temperatures (20°C), the wavelength is in the infrared range centred at 10 μm and the total emitted energy from a perfect blackbody is $0.045\,\mathrm{W\,cm^{-2}}$. Consequently, the infrared emission from warm-blooded animals can be many watts and sensors do not have to be very sophisticated. Many types of infrared detectors have evolved in insects, spiders, reptiles and mammals, each following their own design concept.

The viability of the infrared sensor proposition depends on demonstrating that guard hair structures can achieve the infrared sensitivity required by the imaging function. A typical infrared signal level can be calculated using the radiation laws for a particular scenario. The infrared threat is modelled as a 50 mm diameter source at 1 m range with a temperature of 20°C against a 15°C background. The 5°C differential stems from the author's experience of a typical thermal contrast from the furred areas of warm-blooded

animals against a dry, vegetative background. The antenna collection area is conservatively estimated at 2000 $\mu m^2$. The signal is the difference in infrared power from 20°C and 15°C giving 0.014 nW in a spectral range from 7.5–12.5 µm. In comparison, a good thermal camera under these conditions can achieve a sensitivity of 0.03°C with an F1.2 optic, equivalent to a threshold signal power of 0.01 nW. However, man-made bolometers need to be connected to a silicon readout circuit which can reduce the signal sensitivity and increase excess noise. A better comparison is with infrared sensors from the natural world because the ability to make complex three-dimensional structures can overcome basic sensitivity and noise issues. Fire beetles that are attracted to fires from long distances have been studied in suitable detail. The pyrophilous jewel beetle *Melanophila acuminata* has a thoracic cluster of seventy 12 µm diameter dome-shaped sensilla or photomechanical cells [24]. Each cell has a complex cuticular sphere surrounding a microfluidic core. Most probably this structure provides a dual function of enhancing absorption of infrared energy within the core and controlling the thermal response time. The critical advantage of the insect sensilla is that tiny thermal expansion movements can be detected with a single neuron with stretch-activated ion channels. Sensilla have the potential to be far more sensitive than man-made bolometers. Estimates based on field studies of pyrophilous melanophila beetles have indicated a threshold sensitivity of 0.001 to 0.002 nW [25] or an order more sensitive than man-made bolometers. For the infrared antenna interpretation of guard hair, it can be concluded that the sensitivity requirements are not exceptional, especially when compared with sensors found elsewhere in nature. If non-stochastic processes are employed in the detection process, such as sensing a sudden change in infrared signal, or comparing the response of neighbouring hairs, then the threshold sensitivity is further enhanced.

The infrared antenna concept requires a ring of thermoreceptors at the base of the hair below skin level but above the dermis which absorbs infrared according to its moisture level. The most thorough research on guard hair anatomy has been conducted with emphasis on the tactile function [8]. The guard hair of *Mus musculus* is reported to have a ring of 10 µm diameter Merkel cells around the shaft at the interface of the epidermis and dermis, exactly at the predicted position. Figure 8 shows that these cells are unique to guard hair [8]. The cells are innervated by Aβ-fibres that transmit signals rapidly to the spinal cord and more directly, via the dorsal column nuclei, to the cortex [9]. This is consistent with the rapid response required to an infrared threat. Also, despite protruding from the fur, guard hairs have low tactile sensitivity compared with other hair forms [9]. At the present state of knowledge, the ring of Merkel cells is the main candidate for an infrared thermoreceptor, but it remains speculative until the temperature sensitivity can be tested.

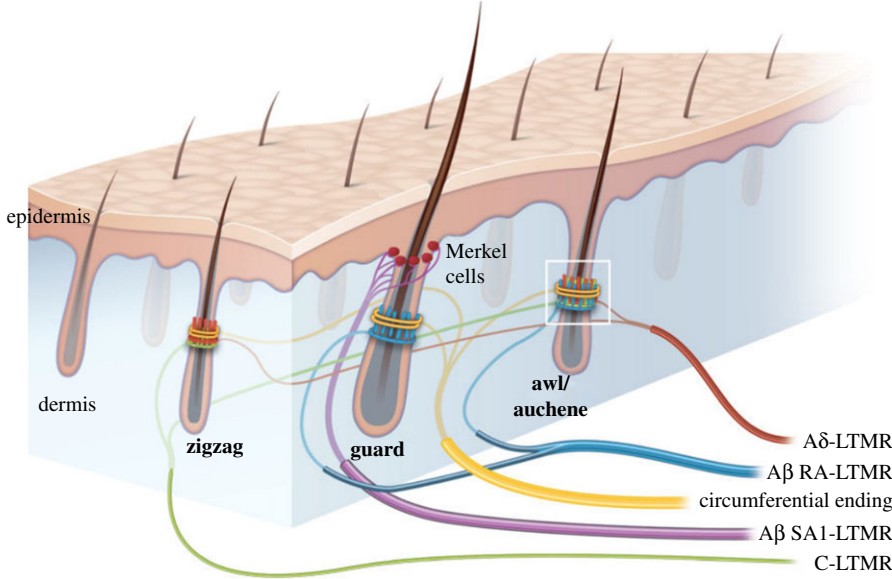

**Figure 8.** Reproduced with permission from 'The gentle touch receptors of mammalian skin' by Amanda Zimmerman, Ling Bai and David D. Ginty of the Department of Neurobiology, Howard Hughes Medical Institute, Harvard Medical School [8]. The illustration shows the complex cell and innervation structure around the base of several mouse hair types including the potential thermoreceptors around the guard hair.

The identification of potential infrared thermoreceptors provides a possible explanation for the very angular and prominent cuticular patterns at the base of the hair where the hair narrows and loses its internal structure (as illustrated in figure 5). As discussed in §3.2, there is evidence that the elongated diamond pattern has a role in inhibiting the radiation losses of the dominant wavelength in tapering sections, effectively acting as a waveguide. As the pattern becomes more prominent towards the base, it is proposed that this topology transforms the infrared photon to a surface wave with most of the energy outside the shaft and directed onto the ring of thermoreceptors.

# 6. Conclusion and further work

The evidence for the infrared antenna interpretation of guard hair is weighted towards the FBG component of the structure described in detail in §3. The banding can be accurately measured and the analysis has allowed mathematical testing of the infrared antenna function. There are no known alternative explanations for the complex micro-structure within these hairs. Rodents, shrews and antechinus have evolved different solutions but all show a precise fit to an infrared antenna tuned to the optimum waveband for thermal imaging between 8 and 12 µm.

The FBGs are simple to measure and analyse in contrast with cuticle patterns that are often quite complex. Establishing a photonic basis for the cuticle pattern is important for the infrared sensor interpretation. The elongated diamond cuticle pattern (illustrated in figure 5 for *Antechinus agilis*) and its role as a wavelength-selective waveguide in tapering sections is our best evidence for the photonic basis. Models are not yet available to handle this level of complexity but may emerge in future. Identifying features in the hair anatomy with a photonic purpose is important for bio-inspiration. Manipulating photons and their polarization state at the scale of the wavelength and in three-dimensions matches the future development path of advanced photonic devices—an important technology for future telecoms, optical computing and sensors. Consequently, there is commercial value in understanding these structures to inspire new man-made devices.

This paper has introduced new analytical tools and concepts specifically for guard hair in small heavily furred animals. The concepts are new and clearly need verifying by other groups. Some key next steps can be highlighted: more expert microscopy, broader species surveys, involvement from the photonics community and well-controlled behavioural studies on wild animals. The tools developed on this programme should enable the structure/function relationships of other hair types to be explained, especially zigzag hair which has a highly specialized topology common across a broad range of species. Guard hair anatomy could support studies on the inter-relatedness of animals, as it appears to be highly resilient over time. Guard hair is the first hair to develop in mice embryos [26], and if this is the evolutionary order, it indicates that the original purpose of hair was infrared protection. Palaeontologists may be able to use this information to explore the evolution of guard hair from its possible emergence in the Triassic period [14].

Ethics. No ethical approvals or licences were required.

Data accessibility. Measurements and more photographs of hair anatomy together with numerical calculations of the filtering function can be found at the Dryad Digital Repository: https://doi.org/10.5061/dryad.np5hqbzrg [21].

Competing interests. The author has no competing interests.

Funding. The author has not received any funding for the research reported.

Acknowledgements. The author wishes to acknowledge the support of Leonardo UK Ltd for use of high-performance thermal imaging cameras and optical microscopes. Thanks to my son Dr David J Baker for valuable advice in preparing the manuscript. Among many contributors I would like to thank Dr Marissa Parrott of Melbourne University, Dr James Gilbert of ANU, Philip Oakley and Kim Lake of Leonardo, Dr Arvind D'Souza of DRS (USA), Professor Mike Benton of Bristol University, Nik Knight, Chairman of Hampshire Bat Group and Peter Whieldon of Otterbourne Wildlife Photographic Centre.

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
