## [Peer Review File · Royal Society Open Science]

Review History

RSOS-210740.R0 (Original submission)

Review form: Reviewer 1

Is the manuscript scientifically sound in its present form?

Yes

Are the interpretations and conclusions justified by the results?

No

Is the language acceptable?

Yes

Do you have any ethical concerns with this paper?

No

Have you any concerns about statistical analyses in this paper?

No

Recommendation?

Accept with minor revision (please list in comments)

Comments to the Author(s)

The questions in the review should be addressed in the manuscript.

Review form: Reviewer 2 (Arvind D'Souza)

Is the manuscript scientifically sound in its present form?

Yes

Are the interpretations and conclusions justified by the results?

Yes

Is the language acceptable?

Yes

Do you have any ethical concerns with this paper?

No

Have you any concerns about statistical analyses in this paper?

No

Recommendation?

Accept as is

Comments to the Author(s)

Please submit all the comments to the editor to the author

Review form: Reviewer 3

Is the manuscript scientifically sound in its present form?

Yes

Are the interpretations and conclusions justified by the results?

Yes

Is the language acceptable?

Yes

Do you have any ethical concerns with this paper?

No

Have you any concerns about statistical analyses in this paper?

No

Recommendation?

Accept with minor revision (please list in comments)

Comments to the Author(s)

The manuscript provides an answer to an interesting question: why is there uniform colour banding along the length of the guard hairs of a variety of small mammals? The author suggests that the banding allows the animal to use its guard hairs as infrared antennae, to detect the

direction and proximity of warm-blooded predators. The author provides a range of pieces of evidence to justify the suggestion, from the fossil record, to the evolutionary divergence of different mammals and marsupials around the world and optical modelling calculations of the periodic structures of the guard hairs. The functional subtlety of the diamond tiling of the scales of the antechinus is not quite explained, but most other features of the guard hair morphology and structure have been plausibly explained.

I think the claim on page 6 line 47 "we know that rodents employ the zipper..." is too strong.

Maybe we propose this? The discussion on page 6 lines 51- 57 is rather speculative.

Another question for the author to consider - does the guard hair enable the small mammal to conserve its own body heat,? (I am not sure about the density of the guard hairs on the body, and whether there is any collective effect of the hair structures?)

Review form: Reviewer 4 (Tim Caro)

Is the manuscript scientifically sound in its present form?

Yes

Are the interpretations and conclusions justified by the results?

No

Is the language acceptable?

Yes

Do you have any ethical concerns with this paper?

No

Have you any concerns about statistical analyses in this paper?

No

Recommendation?

Accept with minor revision (please list in comments)

Comments to the Author(s)

GENERAL

So infrared radiation passes down the hair - but how does it inform the animal that there is something warm nearby - not quite clear to me. There seems to be no discussion of the nervous system involved in sensing the infrared in the epidermis below the hair. We need that evidence dont we?

So this works against terrestrial homeotherms - mammal predators? But are these all ambush predators? I don't think so - certainly not canids that pounce or else run down their prey over long distances. I doubt it works against raptors or owls. So the obvious ambush predators are snakes but these are ectotherms - can you see an infrared signal in such predators? All this needs addressing.

Some pit vipers and possibly other species of snake detect warm blooded prey using infrared. Could these hairs possibly deflect that signal in some way - providing a sort of camouflage?

California ground squirrels signal to snakes that they have seen then using their erect tail as a signal to inform the snake. Could these hairs be involved with such signaling?

It would be helpful to discuss these issues to make the discussion more 'biologically relevant'

SPECIFIC

Abstract: not clear what survival advantage is – spell out here

2,35 no it is not clear that we are missing the structure-function relationships of each one – you need to describe what is known about the functions of each type of hair in more detail before making that statement

2, 40-43 give quantitative measurements of these hairs in different species

Figure 2 is useful but we really need a table with different species' hair measurements side by side for comparison please.

3, 41 not so much missing science but impossible to see how the hair could radiate heat. It will absorb radiation and reradiate it to some extent, depending on the color or reflect it – but I can't see how infra red fits into this well known scenario?

3, 47-53 need literature citations

7, 8 temperature sensitivity of one degree – how can you convince us that this will allow detection of a warm blooded animal from a few meters.

8, 46 radiating photons covering all polarization angles – not clear

Too many figures, drop last two but can you condense them into two or max three?

Interesting idea - don't give up!

Tim Caro

Decision letter (RSOS-210740.R0)

Dear Dr Baker

The Editors assigned to your paper RSOS-210740 "Infrared antenna-like structures in mammalian fur" have now received comments from reviewers and would like you to revise the paper in accordance with the reviewer comments and any comments from the Editors. Please note this decision does not guarantee eventual acceptance.

We do not generally allow multiple rounds of revision so we urge you to make every effort to fully address all of the comments at this stage. If deemed necessary by the Editors, your

manuscript will be sent back to one or more of the original reviewers for assessment. If the original reviewers are not available, we may invite new reviewers.

Please submit your revised manuscript and required files (see below) no later than 21 days from today's (ie 24-Aug-2021) date. Note: the ScholarOne system will 'lock' if submission of the revision is attempted 21 or more days after the deadline. If you do not think you will be able to meet this deadline please contact the editorial office immediately.

on behalf of Dr Jake Socha (Associate Editor) and Pietro Cicuta (Subject Editor)
openscience@royalsociety.org

Associate Editor Comments to Author (Dr Jake Socha):

Associate Editor: 1

Comments to the Author:

Thank you for your patience in awaiting the results of the review. This manuscript required special care, because it proposes a new idea that crosses disciplinary boundaries. The reviewers overall found the IR idea to be interesting, but have numerous comments that need to be addressed in the revised manuscript. In particular, the biological function/mechanism of transduction is not particularly clear, and there needs to be greater discussion in that regard. Merkel cells at the base of the hair are well known to be mechanosensory. So how might the sensing work, biologically? Absorbing IR for thermal insulation is plausible, and has received prior attention. (For example, see Russell, J. E., & Tumlison, R. (1996). Comparison of microstructure of white winter fur and brown summer fur of some arctic mammals. *Acta Zoologica*, 77(4), 279-282.) Please address these issues carefully in your revision.

Reviewer comments to Author:

Reviewer: 1

Comments to the Author(s)

The questions in the review should be addressed in the manuscript (comments below):

Impression:

This manuscript introduces a very interesting concept. The author has done a really good job with a limited resources, he is to be commended for the work. The comments below are basically questions that arouse during the review. The whole concept now needs some experimental work and evidence to support the concept.

General Comments:

1. IR Suppression

What about IR absorption of the prey's heat emission, essentially attenuating the IR emission from the prey, preventing a predator's detection. This seems a more likely applicable function than as a detector, and worth considering. The publication should at least comment on this potential function.

The air cavity in a guard hair may be for dissipation rather than absorption of IR. The author makes a good explanation of the absorption-detection utilization. The dissipation aspect should also be addressed, or at least be ruled out with some calculations or analysis.

2. IR Power

There are expressions that calculate the maximum theoretical power that would be absorbed through the hair's tip or end acting as an antenna. It appears that the manuscript has a good grasp of all the variables needed for such a calculation. Is this theoretical absorbed power level enough to potentially trigger a receptor cell? There is also an efficiency for the theoretical absorption, which is probably difficult to ascertain that would reduce the maximum power.

In Section 4.5, the argument that IR power is converted to skin surface wave at the base of the guard hairs also needs some power analysis to see if enough is available to initiate these waves. Is the power received and transmitted to the hair's base enough to initiate surface waves?

The receptor cells for IR may be tactile sensing cells or some hybrid of these cells. The manuscript should address or speculate on the presence of receptors cells other than the thermoreceptors.

Receptors for this whole concept is a major question for this manuscript! This question needs to have some better calculations to justify its assertion. The outlined innervation and rapid pathway for Merkel cell ring and the low tactile sensitivity is somewhat convincing, but more is needed.

The power available for receptor activation should be addressed. The statement on Page 7 – Line 9 regarding the “limited range of a few meters” should also consider the power available for the range. No explanation for this range statement is given, more is needed.

3. Bio-inspired engineering

If the proposed mechanism were to be proved by experimentation, a great deal of the research would be welcomed, further explored, and potentially utilized.

Specific Comment

Page 9 line 9: innervated no enervated.

Reviewer: 2

Comments to the Author(s)

Please submit all the comments to the editor to the author:

1. Summary of the paper is concise and good
2. Introduction paragraph 1 – Introduced well documented functions of hair but also set up paper for what is not yet known
3. Introduction remainder – Goes from the general properties to the specific setting up for the infrared for mammals from 4 g to 2 Kg a range of 500X in weight
4. Introduction Particularly liked the statement

Page 3 - Lines 38 - 41

5. Section 3 – Sets up infrared basis for the paper – Physics basis is sound for the hypothesis. Shows connection between physics of Bragg grating and mammalian hair – Ties quantitatively the mammalian fur to the Bragg equations
6. Demonstrated in Section 4 that a range of small mammal species have antenna-like features that are tuned to the LWIR 8 – 12 μm range which incorporates the peak in the blackbody radiation for mammalian and “normal” room temperatures in our world

Reviewer: 3

Comments to the Author(s)

The manuscript provides an answer to an interesting question: why is there uniform colour banding along the length of the guard hairs of a variety of small mammals? The author suggests that the banding allows the animal to use its guard hairs as infrared antennae, to detect the direction and proximity of warm-blooded predators. The author provides a range of pieces of evidence to justify the suggestion, from the fossil record, to the evolutionary divergence of different mammals and marsupials around the world and optical modelling calculations of the periodic structures of the guard hairs. The functional subtlety of the diamond tiling of the scales of the antechinus is not quite explained, but most other features of the guard hair morphology and structure have been plausibly explained.

I think the claim on page 6 line 47 "we know that rodents employ the zipper..." is too strong. Maybe we propose this? The discussion on page 6 lines 51- 57 is rather speculative.

Another question for the author to consider - does the guard hair enable the small mammal to conserve its own body heat,? (I am not sure about the density of the guard hairs on the body, and whether there is any collective effect of the hair structures?)

Reviewer: 4

Comments to the Author(s)

GENERAL

So infrared radiation passes down the hair – but how does it inform the animal that there is something warm nearby – not quite clear to me. There seems to be no discussion of the nervous system involved in sensing the infrared in the epidermis below the hair. We need that evidence don't we?

So this works against terrestrial homeotherms – mammal predators? But are these all ambush predators? I don't think so – certainly not canids that pounce or else run down their prey over long distances. I doubt it works against raptors or owls. So the obvious ambush predators are snakes but these are ectotherms – can you see an infrared signal in such predators? All this needs addressing.

Some pit vipers and possibly other species of snake detect warm blooded prey using infrared. Could these hairs possibly deflect that signal in some way – providing a sort of camouflage?

California ground squirrels signal to snakes that they have seen then using their erect tail as a signal to inform the snake. Could these hairs be involved with such signaling?

It would be helpful to discuss these issues to make the discussion more 'biologically relevant'

SPECIFIC

Abstract: not clear what survival advantage is – spell out here

2,35 no it is not clear that we are missing the structure-function relationships of each one – you need to describe what is known about the functions of each type of hair in more detail before making that statement

2, 40-43 give quantitative measurements of these hairs in different species

Figure 2 is useful but we really need a table with different species' hair measurements side by side for comparison please.

3, 41 not so much missing science but impossible to see how the hair could radiate heat. It will absorb radiation and reradiate it to some extent, depending on the color or reflect it – but I can't see how infra red fits into this well known scenario?

3, 47-53 need literature citations

7, 8 temperature sensitivity of one degree – how can you convince us that this will allow detection of a warm blooded animal from a few meters.

8, 46 radiating photons covering all polarization angles – not clear

Too many figures, drop last two but can you condense them into two or max three?

Interesting idea - don't give up!

Tim Caro

===PREPARING YOUR MANUSCRIPT===

If you have been asked to revise the written English in your submission as a condition of publication, you must do so, and you are expected to provide evidence that you have received language editing support. The journal would prefer that you use a professional language editing service and provide a certificate of editing, but a signed letter from a colleague who is a native speaker of English is acceptable. Note the journal has arranged a number of discounts for authors

using professional language editing services
(<https://royalsociety.org/journals/authors/benefits/language-editing/>).

===PREPARING YOUR REVISION IN SCHOLARONE===

<https://royalsociety.org/journals/authors/author-guidelines/#supplementary-material> to include a suitable title and informative caption. An example of appropriate titling and captioning may be found at https://figshare.com/articles/Table_S2_from_Is_there_a_trade-

off_between_peak_performance_and_performance_breadth_across_temperatures_for_aerobic_sc
ope_in_teleost_fishes_/3843624.

Author's Response to Decision Letter for (RSOS-210740.R0)

See Appendix A.

RSOS-210740.R1 (Revision)

Review form: Reviewer 1

Is the manuscript scientifically sound in its present form?

No

Are the interpretations and conclusions justified by the results?

No

Is the language acceptable?

Yes

Do you have any ethical concerns with this paper?

No

Have you any concerns about statistical analyses in this paper?

No

Recommendation?

Accept with minor revision (please list in comments)

Comments to the Author(s)

You should address the theoretical power output, it is a simple calculation and would be a good addition. The explanation of using thermoreceptors is still unbelievable and highly unlikely. It is interesting manuscript and probably deserves publications, however it would be nice if the above items were included and revised.

Review form: Reviewer 3

Is the manuscript scientifically sound in its present form?

Yes

Are the interpretations and conclusions justified by the results?

Yes

Is the language acceptable?

Yes

Do you have any ethical concerns with this paper?

No

Have you any concerns about statistical analyses in this paper?

No

Recommendation?

Accept as is

Comments to the Author(s)

The author has done a careful revision addressing the reviewers' comments. I am happy with the response to my comments.

Decision letter (RSOS-210740.R1)

Dear Dr Baker

The Editors assigned to your paper RSOS-210740.R1 "Infrared antenna-like structures in mammalian fur" have now received comments from reviewers and would like you to revise the paper in accordance with the reviewer comments and any comments from the Editors. Please note this decision does not guarantee eventual acceptance.

Please submit your revised manuscript and required files (see below) no later than 21 days from today's (ie 04-Oct-2021) date. Note: the ScholarOne system will 'lock' if submission of the revision is attempted 21 or more days after the deadline. If you do not think you will be able to meet this deadline please contact the editorial office immediately.

Please note article processing charges apply to papers accepted for publication in Royal Society Open Science (<https://royalsocietypublishing.org/rsos/charges>). Charges will also apply to papers transferred to the journal from other Royal Society Publishing journals, as well as papers submitted as part of our collaboration with the Royal Society of Chemistry

(<https://royalsocietypublishing.org/rsos/chemistry>). Fee waivers are available but must be requested when you submit your revision (<https://royalsocietypublishing.org/rsos/waivers>).

on behalf of Dr Jake Socha (Associate Editor) and Pietro Cicuta (Subject Editor)
openscience@royalsociety.org

Associate Editor Comments to Author (Dr Jake Socha):

Thanks for your revisions to the manuscript. The reviewers are closer to agreement, but there are still two concerns from reviewer 1 that must be addressed. If you need assistance with the power calculation, please let me know.

Reviewer comments to Author:

Reviewer: 1

Comments to the Author(s)

You should address the theoretical power output, it is a simple calculation and would be a good addition. The explanation of using thermoreceptors is still unbelievable and highly unlikely. It is interesting manuscript and probably deserves publications, however it would be nice if the above items were included and revised.

Reviewer: 3

Comments to the Author(s)

The author has done a careful revision addressing the reviewers' comments. I am happy with the response to my comments.

===PREPARING YOUR MANUSCRIPT===

===PREPARING YOUR REVISION IN SCHOLARONE===

<https://royalsociety.org/journals/authors/author-guidelines/#data>. You should ensure that

you cite the dataset in your reference list. If you have deposited data etc in the Dryad repository, please include both the 'For publication' link and 'For review' link at this stage.

Author's Response to Decision Letter for (RSOS-210740.R1)

See Appendix B.

Decision letter (RSOS-210740.R2)

Dear Dr Baker,

It is a pleasure to accept your manuscript entitled "Infrared antenna-like structures in mammalian fur" in its current form for publication in Royal Society Open Science. The comments of the reviewer(s) who reviewed your manuscript are included at the foot of this letter.

The proof of your paper will be available for review using the Royal Society online proofing system and you will receive details of how to access this in the near future from our production

office (openscience_proofs@royalsociety.org). We aim to maintain rapid times to publication after acceptance of your manuscript and we would ask you to please contact both the production office and editorial office if you are likely to be away from e-mail contact to minimise delays to publication. If you are going to be away, please nominate a co-author (if available) to manage the proofing process, and ensure they are copied into your email to the journal.

on behalf of Dr Jake Socha (Associate Editor) and Pietro Cicuta (Subject Editor)
openscience@royalsociety.org

Associate Editor Comments to Author (Dr Jake Socha):

Comments to the Author:

Thank you for the additional revisions – they help to polish this very interesting paper.
Congratulations on your contribution and best wishes!

Appendix A

Comments by reviewers

Associate Editor Comments to Author (Dr Jake Socha):

Associate Editor: 1

Comments to the Author:

Thank you for your patience in awaiting the results of the review. This manuscript required special care, because it proposes a new idea that crosses disciplinary boundaries. The reviewers overall found the IR idea to be interesting, but have numerous comments that need to be addressed in the revised manuscript. In particular, the biological function/mechanism of transduction is not particularly clear, and there needs to be greater discussion in that regard. Merkel cells at the base of the hair are well known to be mechanosensory. So how might the sensing work, biologically? Absorbing IR for thermal insulation is plausible, and has received prior attention. (For example, see Russell, J. E., & Tumlison, R. (1996). Comparison of microstructure of white winter fur and brown summer fur of some arctic mammals. *Acta Zoologica*, 77(4), 279-282.) Please address these issues carefully in your revision.

Extra Section (6) added to address the above:

“Discussion on infrared sensitivity and the potential thermoreceptor”

Reviewer comments to Author:

Reviewer: 1

Comments to the Author(s)

The questions in the review should be addressed in the manuscript (comments below):

Impression:

This manuscript introduces a very interesting concept. The author has done a really good job with a limited resources, he is to be commended for the work. The comments below are basically questions that arouse during the review. The whole concept now needs some experimental work and evidence to support the concept.

General Comments:

1. IR Suppression

What about IR absorption of the prey's heat emission, essentially attenuating the IR emission from the prey, preventing a predator's detection. This seems a more likely applicable function than as a detector, and worth considering. The publication should at least comment on this potential function.

Extra section added (5.2 in green) to explain authors experience

The air cavity in a guard hair may be for dissipation rather than absorption of IR. The author makes a good explanation of the absorption-detection utilization. The dissipation aspect should also be addressed, or at least be ruled out with some calculations or analysis.

The paper concentrates on only one aspect of the function of guard hair. By the laws of thermodynamics absorbing radiators can also act as radiating antennas but the source of infrared is missing in this case

2. IR Power

There are expressions that calculate the maximum theoretical power that would be absorbed through the hair's tip or end acting as an antenna. It appears that the manuscript has a good grasp of all the variables needed for such a calculation. Is this theoretical absorbed power level enough to potentially trigger a receptor cell? There is also an efficiency for the theoretical absorption, which is probably difficult to ascertain that would reduce the maximum power.

Extra section (6 in blue) added to cover this. Unfortunately we do not have the models in photonics to make a quantitative analysis of the efficiency in structures that manipulate photons at the scale of

the wavelength. The assumption is that the antenna is very efficient and this explains why complex filtering is used to squeeze out the last 20% of efficiency.

In Section 4.5, the argument that IR power is converted to skin surface wave at the base of the guard hairs also needs some power analysis to see if enough is available to initiate these waves. Is the power received and transmitted to the hair's base enough to initiate surface waves?

Extra section (6 in blue) added to cover this also

The receptor cells for IR may be tactile sensing cells or some hybrid of these cells. The manuscript should address or speculate on the presence of receptors cells other than the thermoreceptors.

Section 6 is the limit of our current understanding. I understand that the microscopy is very difficult but have added that this should be a key future activity

Receptors for this whole concept is a major question for this manuscript! This question needs to have some better calculations to justify its assertion. The outlined innervation and rapid pathway for Merkel cell ring and the low tactile sensitivity is somewhat convincing, but more is needed.

Hopefully Section 6 expresses this clearly with the caveat that this is unexplored territory

The power available for receptor activation should be addressed. The statement on Page 7 – Line 9 regarding the “limited range of a few meters” should also consider the power available for the range. No explanation for this range statement is given, more is needed.

This section is deleted and replaced by Section 6

3. Bio-inspired engineering

If the proposed mechanism were to be proved by experimentation, a great deal of the research would be welcomed, further explored, and potentially utilized.

Agreed!

Specific Comment

Page 9 line 9: innervated no enervated.

Corrected – thank you

Reviewer: 2

Comments to the Author(s)

Please submit all the comments to the editor to the author:

1. Summary of the paper is concise and good
2. Introduction paragraph 1 – Introduced well documented functions of hair but also set up paper for what is not yet known
3. Introduction remainder – Goes from the general properties to the specific setting up for the infrared for mammals from 4 g to 2 Kg a range of 500X in weight
4. Introduction Particularly liked the statement
Page 3 - Lines 38 - 41

5. Section 3 – Sets up infrared basis for the paper – Physics basis is sound for the hypothesis. Shows connection between physics of Bragg grating and mammalian hair – Ties quantitatively the mammalian fur to the Bragg equations

6. Demonstrated in Section 4 that a range of small mammal species have antenna-like features that are tuned to the LWIR 8 – 12 μm range which incorporates the peak in the blackbody radiation for mammalian and “normal” room temperatures in our world

Reviewer: 3

Comments to the Author(s)

The manuscript provides an answer to an interesting question: why is there uniform colour banding along the length of the guard hairs of a variety of small mammals? The author suggests that the banding allows the animal to use its guard hairs as infrared antennae, to detect the direction and

proximity of warm-blooded predators. The author provides a range of pieces of evidence to justify the suggestion, from the fossil record, to the evolutionary divergence of different mammals and marsupials around the world and optical modelling calculations of the periodic structures of the guard hairs. The functional subtlety of the diamond tiling of the scales of the antechinus is not quite explained, but most other features of the guard hair morphology and structure have been plausibly explained.

I think the claim on page 6 line 47 "we know that rodents employ the zipper..." is too strong. Maybe we propose this? The discussion on page 6 lines 51- 57 is rather speculative.

Sentences rewritten in blue

Another question for the author to consider - does the guard hair enable the small mammal to conserve its own body heat,? (I am not sure about the density of the guard hairs on the body, and whether there is any collective effect of the hair structures?)

The density is low (only 1-3% of the fur) so probably not as strong an insulating component as underfur (zigzag hair – 80%)

Reviewer: 4

Comments to the Author(s)

GENERAL

So infrared radiation passes down the hair – but how does it inform the animal that there is something warm nearby – not quite clear to me. There seems to be no discussion of the nervous system involved in sensing the infrared in the epidermis below the hair. We need that evidence don't we?

New section (6) added to cover this aspect

So this works against terrestrial homeotherms – mammal predators? But are these all ambush predators? I don't think so – certainly not canids that pounce or else run down their prey over long distances. I doubt it works against raptors or owls. So the obvious ambush predators are snakes but these are ectotherms – can you see an infrared signal in such predators? All this needs addressing.

New section (5.2 in green) added to provide Authors experience

Some pit vipers and possibly other species of snake detect warm blooded prey using infrared. Could these hairs possibly deflect that signal in some way – providing a sort of camouflage?

The paper concentrates on infrared sensing in mammals not the suppression of infrared but I am happy to be put in touch with the reviewer to discuss this

California ground squirrels signal to snakes that they have seen them using their erect tail as a signal to inform the snake. Could these hairs be involved with such signaling?

Guard hairs are not very visible for signalling

It would be helpful to discuss these issues to make the discussion more 'biologically relevant'

Have added section 5 to concentrate the biology in one place. There has been a lot of work on filming experiments to test animals for infrared sensitivity but this was excluded from the paper by early reviewers as being uncontrolled.

SPECIFIC

Abstract: not clear what survival advantage is – spell out here

Not appropriate for the Abstract – described in Introduction

2,35 no it is not clear that we are missing the structure-function relationships of each one – you need to describe what is known about the functions of each type of hair in more detail before making that statement

Altered words (in red) in Introduction – Para 1

2, 40-43 give quantitative measurements of these hairs in different species

Figure 1

Figure 2 is useful but we really need a table with different species' hair measurements side by side for comparison please.

Figure 1

3, 41 not so much missing science but impossible to see how the hair could radiate heat. It will absorb radiation and reradiate it to some extent, depending on the color or reflect it – but I can't see how infra red fits into this well known scenario?

This is not the subject of the paper

3, 47-53 need literature citations

There are none – this is being reported for the first time

7, 8 temperature sensitivity of one degree – how can you convince us that this will allow detection of a warm blooded animal from a few meters.

Deleted – replaced with more comprehensive analysis in Section 6

8, 46 radiating photons covering all polarization angles – not clear

Extra clarification added in purple

Too many figures, drop last two but can you condense them into two or max three?

The reviewer wants more information but less figures – can he please be more specific about the structure because this is contradictory

Interesting idea - don't give up!

The paper is a compilation of evidence supporting a structure/function for which there is no current alternative explanation.

Tim Caro

===PREPARING YOUR MANUSCRIPT===

- one version identifying all the changes that have been made (for instance, in coloured highlight, in bold text, or tracked changes);
- a 'clean' version of the new manuscript that incorporates the changes made, but does not highlight them. This version will be used for typesetting if your manuscript is accepted.

If you have been asked to revise the written English in your submission as a condition of publication, you must do so, and you are expected to provide evidence that you have received language editing support. The journal would prefer that you use a professional language editing service and provide a certificate of editing, but a signed letter from a colleague who is a native speaker of English is acceptable. Note the journal has arranged a number of discounts for authors using professional

language editing services (<https://royalsociety.org/journals/authors/benefits/language-editing/>).

===PREPARING YOUR REVISION IN SCHOLARONE===

<https://royalsociety.org/journals/authors/author-guidelines/#supplementary-material> to include a suitable title and informative caption. An example of appropriate titling and captioning may be found at https://figshare.com/articles/Table_S2_from_Is_there_a_trade-

off_between_peak_performance_and_performance_breadth_across_temperatures_for_aerobic_scope_in_teleost_fishes_/3843624.

Appendix B

Final revisions to:

Infrared antenna-like structures in mammalian fur - RSOS-210740

Reviewer: 1

Comments to the Author(s)

You should address the theoretical power output, it is a simple calculation and would be a good addition.

Have added a paragraph:

Thermal infrared radiation is due to a continuous energy interchange between vibrating atoms and electromagnetic waves. At normal ambient temperatures (20°C) the wavelength is in the infrared range centred at 10 microns and the total emitted energy from a perfect blackbody is 0.045 Watts per square centimetre. Consequently, the infrared emission from warm-blooded animals can be many Watts and sensors do not have to be very sophisticated. Many types of infrared detectors have evolved in insects, spiders, reptiles and mammals, each following their own design concept.

The explanation of using thermoreceptors is still unbelievable and highly unlikely.

Have improved the description and added a figure hopefully to make it more believable

The infrared antenna concept requires a ring of thermoreceptors at the base of the hair below skin level but above the dermis which absorbs infrared according to its moisture level. The most thorough research on guard hair anatomy has been conducted with emphasis on the tactile function [8]. The guard hair of *Mus musculus* is reported to have a ring of 10 μm diameter Merkel cells around the shaft at the interface of the epidermis and dermis, exactly at the predicted position. Figure 8 shows that these cells are unique to guard hair [8]. The cells are innervated by $\text{A}\beta$ -fibres that transmit signals rapidly to the spinal cord and more directly, via the dorsal column nuclei, to the cortex [9]. This is consistent with the rapid response required to an infrared threat. Also, despite protruding from the fur, guard hairs have low tactile sensitivity compared with other hair forms [9]. At the present state of knowledge, the ring of Merkel cells is the main candidate for an infrared thermoreceptor, but it remains speculative until the temperature sensitivity can be tested.

Figure 8. Reproduced with permission from “The gentle touch receptors of mammalian skin” by Amanda Zimmerman, Ling Bai, and David D. Ginty of the Department of Neurobiology, Howard Hughes Medical Institute, Harvard Medical School. [8]. The illustration shows the complex cell and innervation structure around the base of several mouse hair-types including the potential thermoreceptors around the guard hair.